# Minimal Inhibitory Concentration (MIC)-Phenomena in *Candida albicans* and Their Impact on the Diagnosis of Antifungal Resistance

**DOI:** 10.3390/jof5030083

**Published:** 2019-09-04

**Authors:** Ulrike Binder, Maria Aigner, Brigitte Risslegger, Caroline Hörtnagl, Cornelia Lass-Flörl, Michaela Lackner

**Affiliations:** Institute of Hygiene and Medical Microbiology, Medical University Innsbruck, 6020 Innsbruck, Austria

**Keywords:** paradoxical effects, trailing, yeasts, in vitro susceptibility testing, minimal inhibitory concentration

## Abstract

Antifungal susceptibility testing (AFST) of clinical isolates is a tool in routine diagnostics to facilitate decision making on optimal antifungal therapy. The minimal inhibitory concentration (MIC)-phenomena (trailing and paradoxical effects (PXE)) observed in AFST complicate the unambiguous and reproducible determination of MICs and the impact of these phenomena on in vivo outcome are not fully understood. We aimed to link the MIC-phenomena with in vivo treatment response using the alternative infection model *Galleria mellonella*. We found that *Candida albicans* strains exhibiting PXE for caspofungin (CAS) had variable treatment outcomes in the *Galleria* model. In contrast, *C. albicans* strains showing trailing for voriconazole failed to respond in vivo. Caspofungin- and voriconazole-susceptible *C. albicans* strains responded to the respective antifungal therapy in vivo. In conclusion, MIC data and subsequent susceptibility interpretation of strains exhibiting PXE and/or trailing should be carried out with caution, as both effects are linked to drug adaptation and treatment response is uncertain to predict.

## 1. Introduction

In times of rising numbers of multi-drug resistant microorganisms, antifungal susceptibility testing (AFST) is an important tool in routine diagnostics to target antifungal therapy. The result of AFST is an antifungal susceptibility profile for the clinical isolate and all systemic antifungal agents. The minimal inhibitory concentrations (MICs) measured for a clinical isolate and subsequent interpretation of the susceptibility based on the clinical breakpoints (CBPs) is one key factor for choosing the most effective therapeutic approach. Other impacting factors for treatment decision making are the location of infection, the underlying disease, and the hepatic and renal function of a patient. Detailed guidance is provided in international guidelines [1]. The two internationally accepted standard AFST methods are EUCAST [2] and CLSI [3]. They allow the categorization of isolates as susceptible, intermediate, or resistant [4] and AFST according to these guidelines has become standard procedure to guide antifungal therapy [5].

For most clinical isolates MIC values can be read without difficulties according to the guidelines [2,3]. However, for some isolates and substances, MIC reading is complicated by the occurrence of MIC-phenomena. MIC-phenomena are defined as the growth effects that hinder the reading of a MIC by clear growth inhibition. Two main phenomena can be distinguished: Trailing and paradoxical effect (PXE). These two phenomena and additional rare phenomena were described in yeasts [6,7,8,9,10,11,12]. Trailing is defined as a reduced turbidity compared to the positive growth control, fully inhibited growth is not achieved and therefore wells fail to become optically clear for MIC reading (as suggested by the guideline) [10]. PXE was defined as reoccurring growth at higher concentrations than the determined MIC. The effects are constantly abounded and vary between species and compounds tested [13], but trailing is occurring mainly for azoles and PXE for echinocandins [10]. The biggest study conducted on the influence of paradoxical effects on the clinical outcome of patients with candidemia was performed by Rueda et al. [10]. They linked the presence of trailing with lower 30-day mortality (odds ratio (OR) 0.55, 95% confidence interval (CI) 0.25-1.00), but not with clinical failure (OR 0.85, 95% CI 0.45-1.54). Although the authors were able to clearly show that in their study PXE and trailing was not associated with clinical failure they advised to be cautious as PXE and trailing is clearly linked with adaptations of the fungi to antifungals and they raised that the phenomena could impact the efficacy of used antifungal agents [10]. The authors highlight that further studies are needed to better interpret the relevance of theses adaptations [10]. Some studies focused on the underlying molecular mechanism of the MIC-phenomena, but the precise underlying mechanisms are not fully understood [14,15,16]. However, upregulation of the chitin synthesis [17,18] and stress response (protein kinase C-, calcineurin-, HOG-, and Hsp90 pathway) is known to play a role in echinocandin PXE [19,20,21]. The histone deacetylase was found to govern the expression of genes related to the early stages of adaptation to azole stress, such as efflux pump genes [22]. Previous studies on MIC-phenomena suggest that some drug-bug-specific effects may correlate with therapeutic failure [16,23,24]. Marr et al. described trailing by *C. albicans* with fluconazole and was able to eliminate trailing by lowering the pH of the medium [25].

Murine models investigating the role of PXE of *Candida* spp. and caspofungin in vivo came to different conclusions. One study concluded that PXE has only limited in vivo significance, because: (a) Survival of animals infected with PXE strains increased under treatment with high CAS concentrations, (b) high fungal load in the kidneys of infected mice receiving 20 mg/kg of CAS was only seen for one strain, and (c) fungal loads could not be reproduced in a repetitive experiment. Bayegan et al. [26] showed, on the other hand, that although survival rates increased in treated groups with most CAS concentrations used, 3 out of 13 mice infected with a *C. tropicalis* strain exhibiting PXE remained infected besides treatment with 2 mg/kg CS. This study concluded that PXE might play a role in vivo for treatment regimens with high CAS doses, and could be a potential source for relapse infections at least for some individuals [16,26]. *G. mellonella* larvae have been used extensively to evaluate the efficacy of antimicrobial drugs [27,28,29,30,31] and to study virulence [32,33,34,35,36,37,38]. Using *Galleria* as a model organism bears plenty of advantages. Larvae are (a) easy to inoculate (due to their size and morphology), (b) can give fast results (24 h–7 days), (c) are cheap to purchase and keep, and (d) no ethical and legal restrictions exist for their use [39]. These lacking restrictions make *Galleria* larvae advantageous for antimicrobial testing, as performance of multiple repetitions and/or increase in sample size, as well as the screening of a large number of different agents is facilitated compared to mammalian models. Additionally, this model was intensively evaluated for its usefulness to investigate antimicrobial efficacy in vivo, e.g., the efficacy of antifungals against infection with *C. albicans* and *C. krusei.* Others evaluated the activity of various antibiotics against *Pseudomonas aeruginosa* and showed that pharmacokinetic parameters correlate well within murine and wax moth models. Importantly, it has been shown that data gained from the wax moth larva correlate well with those obtained in mice [27,31,40,41,42,43,44,45,46].

We aimed to correlate the in vitro observed MIC-phenomena of *C. albicans* with the in vivo treatment response in *G. mellonella* larvae.

## 2. Materials and Methods

### 2.1. Fungal Strains Used in This Study

203 clinical isolates obtained from patients suffering from invasive candidiasis caused by *Candida albicans* were studied during a one year-period at the Division of Hygiene and Medical Microbiology of the Medical University of Innsbruck. Three strains showing MIC-phenomena for caspofungin or voriconazole were chosen for in vivo studies (Table 1). All strains *Candida albicans* strains showing MIC-phenomena (8 out of 203; 3.9%) are given in Appendix A. Strains without MIC-phenomena were selected to prove the accuracy of the *G. mellonella* treatment model (Appendix A).

### 2.2. Antifungal Susceptibility Testing

MIC-phenomena were surveyed as part of the routine antifungal susceptibility testing against voriconazole (VOR), posaconazole (POS), anidulafungin (ANI), and caspofungin (CAS). Strains tested in *Galleria mellonella* were additionally tested against itraconazole (ITR), and fluconazole (FLU).

Yeasts displaying distinctive phenotypical features using E-test (bioMerieux, Vienna, Austria) were further evaluated by microbroth dilution methods according to EUCAST guidelines 9.1. [2]. Briefly, RPMI1640 medium supplemented with 2% glucose was used and inoculum size was adjusted to 1–5 × 10^5^
*Candida* cells/mL. MIC endpoints were determined photometrical after 24 h at 37 °C. Antifungal susceptibility testing was performed in duplicates at independent time points. MICs were defined as the lowest concentration that completely inhibited growth. MICs obtained in trailing phenomena were defined as the concentration showing prominent growth inhibition.

### 2.3. Definition of MIC Phenomena

We focused on strains exhibiting the MIC-phenomena trailing and PXEs. Definitions were in concordance with Rueda et al. [10]. In short, ‘trailing’ was defined as decreasing turbidity when compared to positive control but being constantly present in all wells; the suspension failed to become optically clear [9,47,48]. The ‘PXE’ or eagle-like phenomenon was defined as a growth of organisms at highly elevated concentrations of agents above the apparent MIC [39,49,50]. Species identification was proofed by ITS sequencing. 1.3-beta-d-glucan synthase [51] and 14 alpha-sterol-demethylase [52] of *C. albicans* strains expressing MIC-phenomena were sequenced and analyzed for hotspot mutations according to Lackner et al. [53].

### 2.4. Galleria mellonella In Vivo Treatment Studies

We aimed to evaluate if we can generate a predictive correlation *of* in vitro *and* in vivo results by the use of *G. mellonella* larvae for echinocandins and since no correlation was shown in murine models [11]. As the efficacy of azole therapy in murine models and *Candida* isolates was widely demonstrated in particular for fluconazole [54,55,56,57]; we aimed to investigate the impact of trailing for voriconazole and *C. albicans*.

Accuracy of the model in prediction of CAS-treatment response was tested in a pilot study using *C. albicans* CA 53 (MIC 2 µg/mL for CAS; resistant according to EUCAST CBP; trailing and PXE until 16 µg/mL), and SCH 24 (MIC 0125 µg/mL for CAS; susceptible for CAS according to EUCAST CBP; no MIC phenomena; Table 1). Infected larvae received a single dosage of CAS (0.06 µg/mL, 0.4 µg/mL, and 4 µg/mL), untouched larvae, IPS, and CAS (4 µg/mL) injected larvae served as controls. Accuracy of the model in prediction of the VOR-treatment response was tested in a pilot study using *C. albicans* SCH 39 (MIC 0.0156 µg/mL for VOR; susceptible for VOR according to EUCAST CBP; no MIC phenomena). We were lacking the availability of a *C. albicans* isolate exhibiting resistance, but with clear MIC to VOR, therefore only a susceptible strain without phenomena (SCH 39) was used for the control assays. Infected larvae received a single dosage of VOR (0.06 µg/mL, 16 µg/mL), untouched larvae, insect physiological saline (IPS), and VOR (16 µg/mL) injected larvae served as controls. Experiments were conducted in concordance with the main experiments described in the paragraph below.

Three *C. albicans* strains (SCH 24, SCH 36, and SCH 40) were used for in vivo evaluation in *G. mellonella.* Infection process was performed according a previous study [34]. Sixth-instar larvae of *G. mellonella* (Kurt Pechmann, Langenzersdorf, Austria) were stored in the dark at 18 °C prior to use. Larvae weighing between 0.3 g and 0.4 g were used, each (*n* = 20) infected with 1 × 10^6^ of *C. albicans*. Inocula were diluted in IPS and a volume of 20 µL was injected into the hemocoel via the hind pro-leg. Untouched larvae and larvae injected with 20 µL of IPS or antifungal substance only, served as control. Two hours post-infection, larvae were treated with the respective antifungal agents, incubated at 37 °C in the dark and monitored daily up to 6 days. All larvae were pierced twice, with control groups receiving 20 µL IPS instead of antifungal agents, to rule out an effect on survival curves by the second injection. Concentrations of antifungals were chosen according to in vitro MIC data. Data were evaluated by Kaplan–Meier survival curves utilizing graph pad PRISM software and statistical difference was determined by applying Mantel–Cox log rank test, with *p* ≤ 0.05 regarded as statistically different. All experiments were performed in time-independent duplicates; curves represent average survival of 40 larvae in total.

## 3. Results

*Candida albicans* isolates showed MIC-phenomena for the following azoles FLU, ITR, POS, and VOR (trailing) and for the following echinocandins CAS and ANI (PXE) in our strain collection. MIC-phenomena were never observed for AMB. With an abundance of 3.9% MIC-phenomena in total among *Candida albicans* strains (8 out of 203), trailing and PXE are rare phenomena observed in routine diagnostics (Appendix A). Fifty percent of all strains that were found to exhibit an MIC-phenomena (1.9% of all 203 clinical isolates tested) showed trailing for VOR and/or POS, 50% (1.9% of all 203 clinical isolates tested) showed PXE for CAS and 25% (1.0% of all 203 clinical isolates tested) PXE for ANI (Appendix A).

Fungal strains selected for in vivo studies (Table 1) were tested for the presence or absence of coding mutations within the primary drug target genes that are commonly linked with reduced in vitro susceptibility. With this molecular approach we excluded that the MIC-phenotypes investigated are linked to a resistant genotype associated with hotspot mutations at the primary drug binding site. None of the *C. albicans* strains carried a coding mutation within hot spot regions of lanosterol-14-alpha-demethylase or 1,3-beta-d-glucan-synthase and were for these genes classified as wild-types (data not shown).

Detailed results of pilot experiments are given in Appendix A. As a proof of principle to show that *G. mellonella* is suitable to study treatment outcome, we performed experiments with *C. albicans* SCH 24, CA53, and CAS (SCH 24—susceptible, no phenomenon, CA 53—resistant, and no MIC phenomenon). As expected, survival of larvae infected with the susceptible strain SCH 24 increased upon CAS treatment (*p* = 0.04), while larval survival was not improved for those infected with the resistant isolate (Appendix A). The pilot experiment with VOR showed increased survival of the larvae injected with the VOR susceptible strain SCH 39 and receiving 16 µg of antifungal agent. Although survival in the treated group was higher at all time points after 24 h, differences did not reach statistical significance (*p* = 0.48, log-rank (Mantel–Cox) test).

For testing *C. albicans* strains exhibiting PXEs, the same CAS concentrations, 0.06 µg CAS/larva, mirroring the median MIC of the three chosen test strains, and 4.0 µg CAS/larva, the concentration where persistence started, were used. Increased survival with CAS was detected for isolate SCH 36 (Figure 1A). Here, 50% and 60% of larvae treated with 0.06 µg and 4.0 µg CAS survived in comparison to a survival rate of 20% in the control group. Survival curves were proven to be significantly different (*p* = 0.04) for the higher CAS concentrations administered, but not for the lower concentration (*p* = 0.15) in comparison to the infected, untreated control. SCH 40, the second isolate tested exhibiting the same MIC phenomenon, did not show statistically significant improved survival rates at any of the concentrations (*p* = 0.25 and *p* = 0.71; Figure 1B).

For azole efficacy we assessed either 0.06 µg VOR/larva or 16.0 µg VOR/larva. Again, concentrations were chosen based on the in vitro MIC data. All isolates exhibited a growth inhibition between 0.01–0.06 µg/mL of VOR, but growth persisted up to 16.0 µg/mL. *Galleria* data showed no statistically significant improvement in survival with VOR treatment, independent of the concentration tested and compared to those not receiving VOR (Figure 2). Data obtained assumed that infections with isolates exhibiting trailing phenomenon were unlikely to be successfully treated with VOR, but also treatment of the larvae infected with the susceptible SCH 39, not exhibiting any phenomena, did not result in significant improvement, therefore conclusions need to be taken with care.

## 4. Discussion

### 4.1. MIC-Phenomena

MIC-phenomena are rare with an overall abundancy of 3.9% of all tested *Candida albicans* strains causing candidemia (*n* = 203). At the individual compound level the MIC-phenomena frequency was even lower with 1.9% for VOR, CAS, and POS and 1% for ANI. The study of Marcos-Zambrano et al. [58] found 6.8% trailing for fluconazole and *Candida albicans*. Different to Stevens, we found PXE also for ANI not only for CAS and MICA. Our frequencies for CAS PXE are much lower than the one of the Steven study that reports frequencies of 15% [58]. Since the MIC-phenomena are known to be drug adaptations, the frequencies might not only vary between species and compounds tested, but also between patient cohorts and hospitals. Previous drug exposure of the patient and prophylaxis regimes might also contribute to widely varying frequencies of strains exhibiting MIC-phenomena. The only compound where MIC-phenomena were never observed was AMB, which could provide a treatment alternative for such inconclusive isolates.

We found that our *C. albicans* strains that exhibit MIC-phenomena lacked hot spot mutations in the primary target genes of the antifungal agents (data not shown). In contrast, other studies found that *C. albicans* isolates with strong trailing to fluconazole have the same mutations in ERG11 and up-regulation of efflux pumps as resistant strains [12]. This similar resistance adaptation would suggest that isolates lead to similar therapeutic failures for azole drugs as resistant isolates, however the in vivo murine models suggest that isolates that exhibit trailing respond to azole therapy, which is contractionary [54,55,56,57]. Patient data also suggest that trailing does not confer any advantage to the fungus during the exposure to antifungals in patients. Why there is a lack of correlation between trailing and in vivo resistance is not fully understood, but several hypothesis have been proposed [10]. Among them is that the partial inhibition of growth of the isolates exhibiting trailing might be sufficient for the immune system to control fungal replication in vivo. This could potentially also vary between different patient cohorts and further studies are needed to understand the impact of trailing on drug efficacy.

We know from previous studies that PXE in the presence of echinocandins is the response of yeast adaptation to high antifungal concentrations [11]. Other authors also found that these adaptations are not based on mutations at the primary resistance gene, but by other mechanisms such as cell wall and stress adaptations [19,20,21]. PXE strains show mainly an increase of chitin content that compensates the decrease of β-glucans by antifungal agents [46,50,51]. Other cell stress response such as the protein kinase C-, calcineurin-, HOG-, and Hsp90 pathway that were also found to be involved [19,20,21].

### 4.2. Galleria mellonella as a Model to Study MIC-Phenomena

Herein, we studied the role of MIC-phenomena and evaluated whether these laboratory-based findings imply therapeutic failure using the larvae of the greater wax moth as infection model. We focused on studying the therapeutic response of *C. albicans* infections treated with CAS and VOR as representative agents of the echinocandin and azole class, respectively.

Azoles. The majority of the murine data rely on fluconazole treatment; therefore we aimed to evaluate voriconazole instead as representative compound of the azole drugs in an alternative animal model. First, to generate data also for other azole agents that show in vivo trailing, second, because the compound is well tolerated by the larvae and third as the compound is structurally similar to fluconazole [59]. Accuracy of the *Galleria* model for predicting therapeutic outcome was tested with a VOR susceptible isolate (Appendix A). Consistent, statistically significant data were obtained for *C. albicans* strains that showed in vitro trailing for VOR. *Galleria* survival did not improve under VOR therapy when isolates showed in vitro trailing for VOR (Figure 2). Nevertheless, as also the susceptible control strain lacked statistical significant improvement, although exhibiting higher survival rates in all time points except 24 h, it remains questionable if in vivo predictions of VOR efficacy can be drawn from the *Galleria* model at all. This is contrary to what has been shown for molds, e.g., *A. fumigatus*, for which 1) a clear correlation of the in vitro and in vivo outcome for azole resistant strains was demonstrated and 2) pharmacodynamic analysis revealed the AUC (area under the curve)/MIC ratio determined in hemolymph, a value that is used to predict the outcome based on the stability of the drug in the tissue, was comparable to previous studies, indicating that larvae are trustworthy model systems to test azoles [59].

Echinocandins. Caspofungin was chosen for *G. mellonella* experiments, because it has been used in previous studies exploiting the *Galleria* model and was shown not to be toxic to the larvae and successfully increased survival of *C. albicans* infected larvae [27]. Besides, preliminary assays carried out in our laboratory showed that other echinocandins such as anidulafungin and micafungin seem not to be tolerated by the larvae, as 50% of the control groups died (data not shown, unpublished results).

Murine studies with *Candida* strains exhibiting PXE lack reproducibility [16,26]. Unfortunately, also in the *Galleria* model inconclusive results were found. *Candida albicans* exhibiting PXEs caused variable survival outcome under CAS therapy in *G. mellonella* (Figure 1), whereas strains showing a trailing phenotype for VOR caused a negative therapeutic outcome in *G. mellonella* (Figure 2). In contrast, isolates with definite MICs showed good correlation between MIC results and in vivo outcome (Appendix A), proofing the accuracy of the *G. mellonella* model to predict treatment response for CAS. The variable results for CAS and *Candida albicans* might be also due to a reduced virulence because of a fitness defect of the cells [11]. The impact of compensatory mutations can vary widely between strains. A study assessing the PXE of CAS in therapy of candidiasis in a murine model also showed variable outcome with PXE strains. Here, CAS exhibited efficacy in reducing CFUs in the kidneys, but with no concentration dependency plus CAS could not cure disease. Further, results obtained with one strain could not be reproduced in repetitive experiments [16].

So far, our study shows several limitations as we tested only few strains with MIC-phenomena. Molecular investigations were limited to primary drug target genes; other mechanisms such as efflux pumps and ABC transporter expression of strains expressing PXE and trailing were not studied.

Since the MIC-phenomena were linked with adaptation to antifungal agents, MIC phenomena should be reported as separate category from susceptible and resistant, as response to antifungal treatment might vary, particularly in immunological weakened patient cohorts, as other authors suggested similarly [10].

In conclusion, MIC-phenomena might lead to an underestimation of antifungal resistance, mainly associated with strains that showed trailing for voriconazole, and while the in vivo correlation of the *G. mellonella* infection model and PXE observed for *C. albicans* and CAS were less conclusive, based on the limited number of strains that were assayed in our study. Therefore, MIC data and subsequent susceptibility interpretation should be considered with care.

## Figures and Tables

**Figure 1 jof-05-00083-f001:**
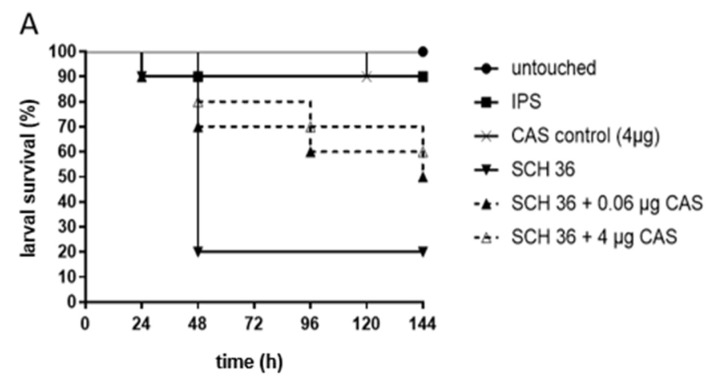
Survival of *G. mellonella* larvae infected with *C. albicans* isolates SCH 36 (**A**) or SCH 40 (**B**) and receiving caspofungin (CAS) treatment. SCH 36 (A) and SCH 40 (B) were shown to exhibit paradoxical growth effects exposed to CAS applying in vitro susceptibility assays. Larvae were infected with 10^6^ cells of the respective isolates and received either 0.06 µg/larva or 4 µg/larva of CAS, 2 h post infections. Insect physiological saline (IPS) was used as an injection control and larvae receiving no antifungals served as control. For detailed information about strains susceptibility profiles see Table 1. Survival curves represent the average survival of all larvae (*n* = 40) tested.

**Figure 2 jof-05-00083-f002:**
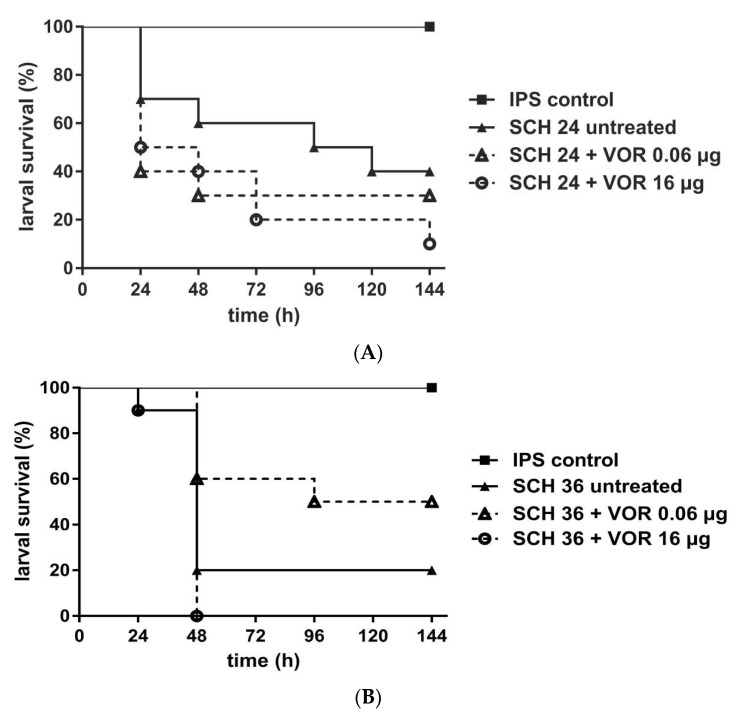
Survival rates of *G. mellonella* larvae infected with *C. albicans* and receiving voriconazole (VOR) treatment. Three different *C. albicans* isolates SCH 24 (**A**), SCH 36 (**B**), or SCH 40 (**C**) were tested, all exhibiting persisting growth with VOR up to 16 µ/mL. Larvae were infected with 10^6^ cells of the respective isolates and received antifungal treatment with 0.06 µg/larva or 16 µg/larva VOR, 2h post infection. Insect physiological saline (IPS) was used as an injection control and larvae receiving no antifungal treatment served as control. For detailed information about strains susceptibility profiles see Table 1. Survival curves represent the average survival of all larvae (*n* = 40) tested.

**Table 1 jof-05-00083-t001:** Overview on selected clinical isolates of *Candida albicans* used for in vivo evaluation in *Galleria mellonella* and their minimal inhibitory concentration (MIC)-phenomena determined by a EUCAST assay.

Strain ID	Species	Antifungal Drug	MIC 24 h (µg/mL)	MIC Phenomena (µg/mL)
SCH 24	*C. albicans*	VOR	0.06	trailing ^a^ > 8
		FLU	0.5	trailing ^a^ > 32
		POS	0.03	trailing ^a^ > 8
		ITR	0.125	trailing ^a^ > 16
		CAS	0.12	
		ANI	0.03	
		AMB	0.25	
SCH 36	*C. albicans*	VOR	0.01	trailing ^a^ > 8
		FLU	1	trailing ^a^ > 32
		POS	0.01	trailing ^a^ > 8
		ITR	0.125	trailing ^a^ > 16
		CAS	0.06	paradoxical growth ^b^ at 4 and 8
		ANI	0.03	
		AMB	0.25	
SCH 40	*C. albicans*	VOR	0.01	trailing ^a^ > 8
		FLU	1	trailing ^a^ > 32
		POS	0.01	trailing ^a^ > 8
		ITR	0.125	trailing ^a^ > 16
		CAS	0.03	paradoxical growth ^b^ at 4 and 8
		ANI	0.03	paradoxical growth ^b^ at 4
		AMB	0.25	
		VOR	0.0156	
		FLU	0.5	
		POS	0.0078	trailing ^a^ > 8
SCH 39	*C. albicans*	ITR	0.125	
		CAS	2	
		ANI	0.06	
		AMB	0.25	
		VOR	0.0156	trailing ^a^ > 8
		FLU	0.5	trailing ^a^ > 32
		POS	0.015	trailing ^a^ > 4
CA 53	*C. albicans*	ITR	0.03	trailing ^a^ > 16
		CAS	2	
		ANI	1	
		AMB	0.25	

VOR, voriconazole; FLU, fluconazole; ITR, itraconazole, POS, posaconazole; CAS, caspofungin; ANI, anidulafungin; AMB amphotericin B; ^a^ trailing is defined as a reduced turbidity compared to the positive growth control, fully inhibited growth is not reacted and therefore will fail to become optically clear for the MIC reading (as suggested by the guideline).; ^b^ paradoxical phenomenon (PXE) was defined as reoccurring growth at higher concentrations than the determined MIC.

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
