# Peer review of "Minimal Inhibitory Concentration (MIC)-Phenomena in Candida albicans and Their Impact on the Diagnosis of Antifungal Resistance"

_jof, 2019, doi:10.3390/jof5030083_

Round 1

Reviewer 1 Report

The objective of this study was to correlate the Minimal Inhibitory Concentration (MIC)-phenomena for C. albicans with in vivo treatment response in Galleria mellonella model. Two MIC-phenomena of the antifungal susceptibility tests were evaluated, including the paradoxical effect (PXE) and trailing. Therefore, the objectives are very interesting and can contribute significantly for the understanding of therapeutic failures in the treatment of candidiasis. Although the study was properly conducted, the text was poorly written. There are many repetitions and confusing parts on the text and the results should be better explored. For publication, many points needs to be improved:

-Abstract: rewrite the results to be clearer;

-Page 2 lines 59-60: complete the phrase with a description of the molecular mechanisms of the MIC-phenomena;

-Page 2 line 66: delete “In the past”;

-Page 2 lines 64-74: This paragraph needs to be restructured: a) explain why murine models fail to generate conclusive results for PXE, b) specify the advantages of Galleria mellonella for antimicrobial tests, and not for infections in general, c) write the objectives in the end of this paragraph;

-Page 2 line 78: Specify if all the clinical isolates (203) were C. albicans?

-Page 2 line 80: The authors mentioned that the strains were randomly selected for the in vivo studies, but in the next pages they explain the reasons to select these strains (lines 168-170);

-Page 2 lines 81-82: Delete “Strains without MIC-phenomena were selected to proof accuracy of the G. mellonella treatment model (Figure S1 and S2)” since this phrase was repeated on the results section;

-Page 4 lines 110-113: The sentence “Hence, of all isolates displaying MIC phenomena, C. albicans (SCH24, SCH36 and SCH40) displaying either trailing or PXE against VOR or CAS were randomly chosen to be evaluated for in vivo outcome in G. mellonella” was repeated more 2 times in the text (page 4 lines 131-133 and Page 5 lines 168-170);

-Page 4 lines 147-151: Do not use “overall” to present the results. More accurate data should be added about the results of MIC-phenomena. I suggest an addition of a Table containing the percentages of PXE and trailing for each antifungal found among the 203 isolates;

-Page 5 lines 175-176: “Survival curves were proven to be significantly different (p=0.04) for the higher CAS concentrations administered”. The survival was significantly different in relation to the control group not-treated infected? It is not clear;

-In the results section, the authors should present the results obtained in the assays of PXE for ANI, and trailing for FLU, POS e ITR, including graphs and statistical analysis;

-The discussion section needs to be restructured with addition of paragraphs and a better organization of the topics;

-Page 8 lines 268-274: This paragraph is out of the context;

-Page 8 lines 275-281: The conclusion “The only compound where MIC trailing was never observed was AMB, which could provide a treatment alternative for such inconclusive isolates” not correspond with the objectives of this study. It should be mentioned only in the discussion.

Author Response

Reviewer 1

The objective of this study was to correlate the Minimal Inhibitory Concentration (MIC)-phenomena for C. albicans with in vivo treatment response in Galleria mellonella model. Two MIC-phenomena of the antifungal susceptibility tests were evaluated, including the paradoxical effect (PXE) and trailing. Therefore, the objectives are very interesting and can contribute significantly for the understanding of therapeutic failures in the treatment of candidiasis. Although the study was properly conducted, the text was poorly written. There are many repetitions and confusing parts on the text and the results should be better explored. For publication, many points needs to be improved:

We rewrote the paragraph.

-Abstract: rewrite the results to be clearer;

We rewrote the abstract.

-Page 2 lines 59-60: complete the phrase with a description of the molecular mechanisms of the MIC-phenomena;

The known molecular mechanisms are now described.

-Page 2 line 66: delete “In the past”;

Done.

-Page 2 lines 64-74: This paragraph needs to be restructured: a) explain why murine models fail to generate conclusive results for PXE, b) specify the advantages of Galleria mellonella for antimicrobial tests, and not for infections in general, c) write the objectives in the end of this paragraph;

We have restructured and rewritten this paragraph accordingly.

-Page 2 line 78: Specify if all the clinical isolates (203) were C. albicans?

Yes, all isolates were C. albicans.

-Page 2 line 80: The authors mentioned that the strains were randomly selected for the in vivo studies, but in the next pages they explain the reasons to select these strains (lines 168-170);

Strains for the in vivo experiments were selected randomly (5 out of 8), but for the pilot experiments strains with distinct different MIC phenomena were selected.

-Page 2 lines 81-82: Delete “Strains without MIC-phenomena were selected to proof accuracy of the G. mellonella treatment model (Figure S1 and S2)” since this phrase was repeated on the results section;

Line was deleted.

-Page 4 lines 110-113: The sentence “Hence, of all isolates displaying MIC phenomena, C. albicans (SCH24, SCH36 and SCH40) displaying either trailing or PXE against VOR or CAS were randomly chosen to be evaluated for in vivo outcome in G. mellonella” was repeated more 2 times in the text (page 4 lines 131-133 and Page 5 lines 168-170);

               Redundant phrases were deleted.

-Page 4 lines 147-151: Do not use “overall” to present the results. More accurate data should be added about the results of MIC-phenomena. I suggest an addition of a Table containing the percentages of PXE and trailing for each antifungal found among the 203 isolates;

A table with MICs and effects per compound were added in the supplementary materials.

-Page 5 lines 175-176: “Survival curves were proven to be significantly different (p=0.04) for the higher CAS concentrations administered”. The survival was significantly different in relation to the control group not-treated infected? It is not clear;

The CAS curves are significantly different in comparison to the untreated control group: infected but without receiving caspofungin. This information was added to the text.-

-In the results section, the authors should present the results obtained in the assays of PXE for ANI, and trailing for FLU, POS e ITR, including graphs and statistical analysis;

The comment of the reviewer is not clear. Only 3.9% of strains (total 203) showed trailing or PXE. In numbers these are 8 strains that are given in the supplementary materials in a separate table.

We don’t know which kind of graphic or statistical analysis the editor has in mind.

-The discussion section needs to be restructured with addition of paragraphs and a better organization of the topics;

               We rewrote the discussion to increase readability.

-Page 8 lines 268-274: This paragraph is out of the context;

We agree and have deleted this paragraph here in the discussion. Detailed information on the value of the larval model for antimicrobial efficacy testing has been added to the introduction (see comment above), and by deleting this paragraph here, we omit further repetitions.

-Page 8 lines 275-281: The conclusion “The only compound where MIC trailing was never observed was AMB, which could provide a treatment alternative for such inconclusive isolates” not correspond with the objectives of this study. It should be mentioned only in the discussion.

Was shifted to discussion.

Reviewer 2 Report

General comments

I have reviewed this manuscript with interest which focused on the MIC-phenomena and their impacts. The MIC-phenomena included training and paradoxical effects, two important phenomena which may interfere with the determination of the minimum inhibitory concentrations.

Specific comments

Abstract.

Line 11, replace "decide on" with "facilitate making the decision on selection optimal antifungal therapy".

Line 13, replace "correct diagnosis of MIC reading" with "unambiguous and reproducible determination of MICs".

Line 15, replace "Candida (C.) albicans" with "Candida albicans".

Introduction

Line32, replace “minimal inhibitory concentrations” with “minimal inhibitory concentrations (MIC)”

Line 67, may consider replace “pathogenic” with either “virulence” or “pathogenesis”.

Line 103, “where trailing and PXEs” seem incomplete sentence. Please clarify.

Line 120-121, “trailing and PXE unitl/at 16 ug/mL” seem typo, please clarify.

Line 121, SCH24 “MIC 0.25 for CAS” not consistent with data in Table 1 (CAS 0.12). Please clarify.

Line 123, please replace “IPS” with “insect physiological saline (IPS)”.

Line 124-125, SCH39 (MIC 0.0078 for VOR” not consistent with data in Table 1 (VOR 0.0156). Please clarify.

Line 135, replace “insect physiological saline (IPS)” with “IPS”.

Discussion

Suggest split the first paragraph into two paragraphs, one focusing on trailing and paradoxical, the other focusing on model.

Author Response

General comments

I have reviewed this manuscript with interest which focused on the MIC-phenomena and their impacts. The MIC-phenomena included training and paradoxical effects, two important phenomena which may interfere with the determination of the minimum inhibitory concentrations.

Specific comments

Abstract.

Line 11, replace "decide on" with "facilitate making the decision on selection optimal antifungal therapy".

Done.

Line 13, replace "correct diagnosis of MIC reading" with "unambiguous and reproducible determination of MICs".

Done.

Line 15, replace "Candida (C.) albicans" with "Candida albicans".

Done.

Introduction

Line32, replace “minimal inhibitory concentrations” with “minimal inhibitory concentrations (MIC)”

Done.

Line 67, may consider replace “pathogenic” with either “virulence” or “pathogenesis”.

Done.

Line 103, “where trailing and PXEs” seem incomplete sentence. Please clarify.

Done.

Line 120-121, “trailing and PXE unitl/at 16 ug/mL” seem typo, please clarify.

 Done.

Line 121, SCH24 “MIC 0.25 for CAS” not consistent with data in Table 1 (CAS 0.12). Please clarify.

 Line 123, please replace “IPS” with “insect physiological saline (IPS)”.

 Done.

Line 124-125, SCH39 (MIC 0.0078 for VOR” not consistent with data in Table 1 (VOR 0.0156). Please clarify.

            Done.

Line 135, replace “insect physiological saline (IPS)” with “IPS”.

            Done.

Discussion

Suggest split the first paragraph into two paragraphs, one focusing on trailing and paradoxical, the other focusing on model.

Done.

Round 2

Reviewer 1 Report

Most of comments were addressed. However, a minor review is still necessary:

-Abstract line 12: Replace  “The minimal inhibitory concentration (MIC)-phenomena (trailing and paradoxical effects (PXE)) observed in AFST…” by “The minimal inhibitory concentration (MIC)-phenomena, trailing and paradoxical effects (PXE), observed in AFST…”

-Lines 80-91: The phrases “Additionally, this model was intensively evaluated for its usefulness to investigate antimicrobial efficacy in vivo,e.g., the efficacy of antifungals against infection with C. albicansand C. krusei. Others evaluated the activity of various antibiotics against Pseudomonas aeruginosaand showed that pharmacokinetic parameters correlate well within murine and wax moth models” are repeated in the next paragraph. Although the phrases are the same, some references cited are different.  

-Line 172: Write Table S1 only in the end of the paragraph.

-The Table 1 and Table S1 are very similar and repeat most of results. Since Table S1 is more complete, I suggest use only this one. Transfer the Table S1 for Table 1.

Author Response

Most of comments were addressed. However, a minor review is still necessary:

-Abstract line 12: Replace  “The minimal inhibitory concentration (MIC)-phenomena (trailing and paradoxical effects (PXE)) observed in AFST…” by “The minimal inhibitory concentration (MIC)-phenomena, trailing and paradoxical effects (PXE), observed in AFST…”

Response: We changed it accordingly. 

-Lines 80-91: The phrases “Additionally, this model was intensively evaluated for its usefulness to investigate antimicrobial efficacy in vivo,e.g., the efficacy of antifungals against infection with C. albicansand C. krusei. Others evaluated the activity of various antibiotics against Pseudomonas aeruginosaand showed that pharmacokinetic parameters correlate well within murine and wax moth models” are repeated in the next paragraph. Although the phrases are the same, some references cited are different. 

Response: We correct this. 

-Line 172: Write Table S1 only in the end of the paragraph.

-The Table 1 and Table S1 are very similar and repeat most of results. Since Table S1 is more complete, I suggest use only this one. Transfer the Table S1 for Table 1.

Response: The datasets of Table 1 and Table S1 are not the same. Table 1 contains all strains that were tested in the Galleria model including the test strains that were used as controls. While in Table S1 all eight strains  strains are presented. Merging the tables would lead to confusion.